# Targeting Translation and the Cell Cycle Inversely Affects CTC Metabolism but Not Metastasis

**DOI:** 10.3390/cancers15215263

**Published:** 2023-11-02

**Authors:** Tetiana Y. Bowley, Seth D. Merkley, Irina V. Lagutina, Mireya C. Ortiz, Margaret Lee, Bernard Tawfik, Dario Marchetti

**Affiliations:** 1Division of Molecular Medicine, Department of Internal Medicine, University of New Mexico Health Sciences Center, Albuquerque, NM 87131, USA; tbowley@salud.unm.edu (T.Y.B.); smerkley@salud.unm.edu (S.D.M.); mcortiz@salud.unm.edu (M.C.O.); marlee@salud.unm.edu (M.L.); 2Animal Models Shared Resource, University of New Mexico Comprehensive Cancer Center, Albuquerque, NM 87120, USA; ivlagutina@salud.unm.edu; 3Division of Hematology and Oncology, Department of Internal Medicine, University of New Mexico Comprehensive Cancer Center, Albuquerque, NM 87120, USA; btawfik@salud.unm.edu

**Keywords:** circulating tumor cells (CTCs), metastatic melanoma, omacetaxine, palbociclib, cell metabolism, translational inhibition, cell cycle inhibition

## Abstract

**Simple Summary:**

This study identifies specific ribosomal proteins that drive MBM and extracranial metastasis via CTCs. Dual targeting of cellular translation and proliferation prevents metastasis of aggressive CTC subsets with high RPL/RPS expression. We report the first-ever real-time metabolic flux analysis of patient-derived melanoma CTCs and altered carbohydrate metabolism during impaired translation in a melanoma-CTC-derived clone.

**Abstract:**

Melanoma brain metastasis (MBM) is significantly associated with poor prognosis and is diagnosed in 80% of patients at autopsy. Circulating tumor cells (CTCs) are “seeds” of metastasis and the smallest functional units of cancer. Our multilevel approach has previously identified a CTC RPL/RPS gene signature directly linked to MBM onset. We hypothesized that targeting ribogenesis prevents MBM/metastasis in CTC-derived xenografts. We treated parallel cohorts of MBM mice with FDA-approved protein translation inhibitor omacetaxine with or without CDK4/CDK6 inhibitor palbociclib, and monitored metastatic development and cell proliferation. Necropsies and IVIS imaging showed decreased MBM/extracranial metastasis in drug-treated mice, and RNA-Seq on mouse-blood-derived CTCs revealed downregulation of four RPL/RPS genes. However, mitochondrial stress tests and RT-qPCR showed that omacetaxine and palbociclib inversely affected glycolytic metabolism, demonstrating that dual targeting of cell translation/proliferation is critical to suppress plasticity in metastasis-competent CTCs. Equally relevant, we provide the first-ever functional metabolic characterization of patient-derived circulating neoplastic cells/CTCs.

## 1. Introduction

Melanoma is the most aggressive form of skin cancer with poor therapeutic options and clinical outcomes [1,2,3]. Metastatic dissemination to the brain occurs in 60% of advanced melanoma patients and in up to 80% of patients at autopsy [1,4]. Patients with melanoma brain metastasis (MBM) have a life expectancy of only 4–6 months, associated with a rapid decrease in life quality [1,2,5,6]. Localized therapies include surgical removal and radiation, while metastatic melanoma is effectively treated with immunotherapy and targeted agents [1,7,8]. Although innovative therapeutic breakthroughs are constantly developed, substantial advances remain elusive and most MBM patients die due to MBM progression, hemorrhage, and intra- and extracranial metastases [9,10].

Metastatic dissemination to the brain and other organ sites occurs due to continuous shedding of circulating tumor cells (CTCs) from primary and/or metastatic tumors into the bloodstream [2,11,12,13]. CTCs are “seeds” of fatal metastasis and the smallest functional units of cancer. Hypoxia is one of the major factors that triggers CTC shedding and enhanced metastasis [14,15]. Due to the detrimental effects of shear stress in blood, CTC clusters are dissociated and the vast majority of CTCs die prior to successful dissemination to distant organs [14,16,17,18,19]. CTCs that do disseminate can remain dormant for years, e.g., in the bone marrow as depot, or initiate clinically undetectable micrometastases [20,21,22]. CTCs employ diverse invasive behaviors, such as amoeboid cellular invasion, CTC cluster streaming, or collective cell migration and invasion [14]. The latter is the major CTC behavioral method for establishment and progression of a solid tumor [14]. Significantly, high numbers of single CTCs and/or CTC clusters are directly associated with cancer severity and clinical outcomes [12,23,24,25,26]. CTC clusters are composed of leader and follower cells, with leaders preparing a low-resistance path to shield followers from biomechanical stressors in the blood such as shear stress, tension, and compression [14,27]. By traveling with companions, including immune and stromal cells, CTCs can increase their survival chances in the harsh microenvironment of blood [14]. We have previously reported that growing MBM induces exponential shedding of CTCs into the bloodstream [28].

We employed an unbiased multilevel approach to identify a unique 21-member RPL/RPS signature of melanoma CTCs that drives MBM onset and progression (“RPL” means proteins of the large ribosomal 60S subunit, while “RPS” refers to proteins of the small ribosomal 40S subunit [28]). Translational reprogramming of cancer cells occurs early on in tumorigenesis to initiate tumor growth, dissemination of CTCs into the bloodstream, and metastasis [29]. Cancer cells can rewire ribogenesis to preferentially translate oncogenic transcripts and promote metastatic spread, neoplastic growth, and survival, and this deregulation has been linked to increased metastatic spread of cancer [29,30,31,32]. For example, upregulation of RPL15 gene expression in breast cancer promotes massive metastasis and triggers enhanced translation of other ribosomal proteins [29,30], while inhibition of RPL27A hampers dissemination and invasive potential of breast cancer cells [33].

In synchrony with the above studies, we have identified 21 RPL/RPS genes significantly upregulated in CTCs that drive MBM onset [28]. Because there is highly regulated cooperation between ribosome assembly and cell-cycle progression, and since inhibition of translation blocks cell proliferation [31,34], we hypothesized that targeting ribosome biogenesis and/or cell proliferation prevents MBM development in CTC-derived xenografts (CDXs). We evaluated this by inhibiting these two vital processes in MBM CDXs using the translational inhibitor omacetaxine and/or the cell-cycle inhibitor palbociclib. Omacetaxine inhibits peptide elongation by directly binding to the A-site cleft of the ribosome and, while the exact mechanism of ribosomal inhibition remains to be elucidated [35], it is FDA-approved for treating chronic myelogenous leukemia [36,37]. Conversely, palbociclib is a CDK4/6 inhibitor that has been widely used in many cancer therapies [38]. Here, we demonstrate that targeting either the cell translational machinery and/or cell proliferation results in a significant decrease in MBM/metastasis. Additionally, we singled out 4 key players from the 21-member CTC RPL/RPS gene signature as significantly downregulated following drug treatment and revealed carbohydrate metabolism to be dysregulated during impaired ribogenesis. Metabolic rewiring is fundamental to metastatic progression and metastasis-competent CTCs are hypothesized to be capable of flexibly deploying both oxidative and anabolic metabolism to meet energy demands, overcome oxidative stress, and expand available metastatic niches [39,40,41,42,43,44]. We report the first-ever metabolic flux analysis of patient-derived neoplastic cell fractions/CTCs in real-time, alongside omacetaxine and/or palbociclib treatment of a melanoma CTC-derived clone to elucidate melanoma CTC metabolism during inhibition of ribogenesis.

## 2. Materials and Methods

### 2.1. Patients’ Blood Collection

Metastatic melanoma patient samples were accrued according to protocols approved by the Institutional Review Board at UNM Health Sciences Center (UNM-HSC) in Albuquerque, New Mexico. Patients’ clinical parameters are shown in Appendix A. Patient samples were collected after completed informed written consent was obtained. Under strict aseptic conditions, peripheral blood (12–18 mL) from each patient was collected at the middle of vein puncture in a sodium-etheylenediamine tetraacetic acid (EDTA) tube as part of their doctor’s visit. Blood samples were provided immediately to the laboratory after collection for CTC isolation and examination, and processed immediately or fixed in a CellSave^TM^ tube (Menarini Silicon Biosystems, Inc., Bologna, Italy).

### 2.2. CTC-Derived Xenografts (CDXs)

All in vivo studies were approved by Institutional Animal Care and Use Committee (IACUC) protocol. The generation of CTC-derived xenografts (CDXs) was executed using 6 to 8 week old NBSGW (NOD.Cg-KitW-41J Tyr + Prkdcscid Il2rgtm1Wjl/ThomJ) immunodeficient mice. Thirty mice received 50 µL (4 mg/mL) of low-molecular weight heparin (NDC, Cat# 63323-533-01, Lake Zurich, IL, USA) retro-orbitally 10 min before intracardiac injection of CTC-derived CTC-Luc2 clone to decrease thromboembolism-related mortality in mice [45]. Following anesthesia with isoflurane (2.5%, 1 L/min O_2_ flow), animals were injected under aseptic conditions into the left ventricle with 5.0 × 10^5^ cells in 50 µL of PBS using a sterile 0.5 mL U-100 insulin syringe with a 29G×/2″ needle (Beckton Dickinson, Cat# 58324702, Franklin Lakes, NJ, USA). Successful intracardiac injection was confirmed by blood backflow into the syringe prior to cell injection. MBM development was confirmed as early as 24 h post-injection using the Xenogen IVIS Spectrum animal imager (PerkinElmer, Waltham, MA, USA) following intraperitoneal injection of D-luciferin (Sigma, Cat# 2591-17-5; 150 mg/kg, St. Louis, MO, USA). IVIS imaging was performed on weekly basis to monitor tumor development. One week post-injection, 20 MBM mice were equally divided between four cohorts: (1) placebo mice which received vehicle (*n* = 5); (2) omacetaxine mice which received IP injection of omacetaxine (Chem Scene, Cat# CS-2872; 0.5 mg/kg, Monmouth Junction, NJ, USA), *n* = 5; (3) palbociclib mice which received oral gavage of the palbociclib (Selleckchem, PD-0332991; 25 mg/kg, Houston, TX, USA), *n* = 5; (4) mice which received combined treatment, *n* = 5. Drug treatment was performed every weekday, as previously described, over a period of three weeks [38,46]. After daily monitoring of animals over the period of four weeks for changes in overall health status (sudden weight loss, rough coat, distress, difficulty with ambulation, seizures, or difficulty in obtaining food or water), all the animals were sacrificed for subsequent necropsies. Approximately 800 to 900 µL of mouse blood was collected into an EDTA-containing MiniCollect tube (Greiner Bio-One, Cat# K3E K3EDTA, Monroe, NC, USA). Brain, liver, lungs, spleen, sternum, and skull-cap tissues were fixed in 10% neutral buffered formalin for further pathological assessment. Since most melanoma cells produce melanin, melanoma macrometastases were visually identified as brown-to-black pigmented areas [47].

### 2.3. CTC/CTC Cluster Capture

Mouse blood (100–150 µL) was collected retro-orbitally using EDTA-coated glass Pasteur pipette into a MiniCollect tube (Greiner Bio-One, Cat# K3E K3EDTA, Monroe, NC, USA) and loaded onto the CTC Parsortix^TM^ microfluidic chip (8 µm) within 1 h of mouse necropsies. CTC and CTC clusters were captured by the filtration/microfluidic Parsortix PR1 platform (Angle Europe Ltd., Surrey, UK) using 6.5 µM cartridges (Angle, PLC). For immunofluorescence staining, CTCs were captured by Parsortix based upon their size and deformability, and subjected to on-cassette staining, as previously described [25,28]. CTCs were designated as cells positive for human Mel-A (Alexa Fluor 594-tagged, Santa Cruz Biotechnology, Cat#sc-20032, Dallas, TX, USA) and human DAPI (Thermo Fisher, Cat# D3571, Waltham, MA, USA) staining, but negative for human CD45 (FITC-tagged, BioLegend, Cat#103108, San Diego, CA, USA). CTC visualization and quantification were performed using Zeiss LSM800 microscope (10–40× magnification) and ZEN Blue 2.1 software (Carl Zeiss Microscopy; Oberkochen, Germany). For RNA-Seq analyses, CTCs were captured and harvested using Parsortix, according to manufacturer’s guidelines.

### 2.4. RNA Sequencing

RNA was isolated from enriched and flow-through fractions (10–20 × 10^3^ cells) after Parsortix^TM^ CTC harvesting (ANGLE, LLC, Plymouth Meeting, PA, USA). Enriched fraction contained human CTCs, while flow-through fraction consisted of immune cells. RNA analysis, cDNA synthesis, and library preparation were performed using the microarray platform SMARTer Universal Low Input RNA kit for RNA sequencing (Clontech, Cat#634946, San Jose, CA, USA). Fragmented RNA was processed using the Ion Plus Fragment Library kit (Thermo Fisher, Cat#4471252, Waltham, MA, USA), as previously reported [48,49]. The samples were sequenced using the Ion Proton S5/XL platform (Thermo Fisher, Waltham, MA, USA) at the Analytical and Translational Genomics (ATG) Shared Resource Core at the University of New Mexico Comprehensive Cancer Center (UNM-CCC).

### 2.5. RNA-Seq Analyses and Bioinformatics

Sequence alignment was performed using tmap (v5.10.11) to a BED file carrying non-overlapping exon regions from the UCSC genome browser (GRCh38/hg38). Exons were quantified using the HTSeq (v0.11.1) Python platform [50,51]. Generation of gene counts was performed by summing counts across exons. Library size was normalized and differential analysis was executed using edgeR [50,52]. An adjusted *p*-value of cutoff of 0.05 was used for analyzing edgeR and DESeq values. Heatmap3 was used to generate heatmaps and execute cluster analyses. Pathway enrichment analyses were conducted using clusterProfiler, GAGE, Pathview, and topGO software programs, R Package Version 2.42.0 [50,52]. The Reactome pathway database was used to generate pathway discrimination analyses [53]. The crystal structure of the ribosome and its components was generated using the PyMOL molecular visualization system, with the specific structure code being 4UG0 [54]. The relevant proteins were colored using PyMOL and the individually referenced proteins within the protein databank [55].

### 2.6. Cell Culture

Early-passage brain-metastatic melanoma CTC-derived clonal cell line (CTC clone; generated in Dr. Marchetti’s laboratory) [28,56] or human lung-metastatic melanoma MeWo cell line (ATCC, Cat# HTB-65, Manassas, VA, USA) were cultured in DMEM/F12 (Thermo Fisher, Cat# 11320082), supplemented with 10% fetal bovine serum (FBS) (Gibco, Cat# A4766801, Billings, MT, USA) and 1X GlutaMAX (Gibco, Cat# 35050-061), termed “standard media” (Appendix A). Cells were maintained at 37 °C in a humidified 5% CO_2_ incubator and passaged using 0.05% Trypsin-EDTA (Gibco, Cat#** 25300054) before reaching confluency. Routine testing for Mycoplasma was performed using the Mycoplasma Detection Assay (MycoAlert, Lonza, Morristown, NJ, USA) every 20 passages. Cell-line authentication was conducted using PCR-based assay routinely. Luciferase-tagged CTC clone cells were generated as previously reported [57]. Prior to injection into mice, cells were checked for morphological and phenotypic changes using microscopy.

### 2.7. Blood Separation and Lineage Sorting

Whole blood obtained as above was subjected to magnetic labeling for erythrocyte and granulocyte depletion per the manufacturer’s instructions using the Custom Whole Blood PBMC Iso Kit (Miltenyi Cat# 130-126-359, Auburn, CA, USA), autoMACS Running Buffer (Miltenyi, Cat# 130-091-221), and Custom Sedimentation Kit2 (Miltenyi, Cat# 130-126-357). Lineage sorting was performed using LS Positive Selection Columns (Miltenyi, Cat# 130-126-357), anti-CD45 microbeads (Miltenyi, Cat# 130-045-801), and the QuadroMACs Magnetic Separator (Miltenyi, Cat# 130-091-051). After granulocyte and erythrocyte depletion, cells were washed in autoMACS buffer, and re-suspended in 300 µL of autoMACS and 100 µL of Fc Block (BioLegend, Cat# 422302, San Diego, CA, USA), each per 5 × 10^7^ cells, and incubated at 4 °C for 10 min. An amount of 100 µL of CD45 microbeads per 5 × 10^7^ cells were added prior to 30 min incubation at 4 °C. At 15 min into the 30 min incubation, 100 µL of Dead Cell Removal Beads (Miltenyi, Cat# 130-090-101) per 1 × 10^7^ cells was added. Fractions were then subject to magnetic separation in LS Positive Selection Columns and washed with PBS, and the CD45− and CD45+ (Lin− and Lin+, respectively) fractions were then used for metabolic flux analysis.

### 2.8. Metabolic Flux Analysis on Patient-Derived PBMCs and CTC-Derived Clone

Mitochondrial stress tests (MSTs) (Agilent, Cat# 103015-100, Santa Clara, CA, USA) and Glycolytic Rate Assays (GRAs) (Agilent, Cat# 103344-100) were performed according to the manufacturer’s instructions. Assay medium for all experiments was XF DMEM, pH 7.4 (Agilent, Cat# 103575-100), supplemented with 10 mM glucose (Agilent, Cat# 103577-100), 1 mM pyruvate (Agilent, Cat# 103578-100), and 2 mM glutamine (Agilent, Cat# 103579-100). MST port drugs were used at final well concentrations as follows: oligomycin (3 µM), FCCP (3 µM) [58,59], and rotenone/antimycin A (Rot/AA, 1 µM) [60]. GRA port drugs were 2-deoxyglucose (2-DG, 5 µM) and Rot/AA (500 mM). Post-assay normalization on suspension cells was performed by including Hoechst Dye (Life Technologies, Cat# H3570) with the final drug injection of the Seahorse^TM^ assay (Agilent, Inc.), yielding a final well concentration of 16 µM. Plates were then analyzed using a BioTek Cytation 1 Cell Imaging Multimode Reader running Gen5 3.11 Software programmed to output MeanTsf values, which were multiplied by input viability to yield the final normalization unit for PBMC assays. Normalization was performed using Agilent Wave 2.6.1.56 Software. Prior to use, Seahorse Cell Culture Plates (Agilent, Cat# 102416-100) were coated overnight in a Biosafety Cabinet with 100 µL per well of 100 µg/mL PureCol-S Solution (ECM Protein, Cat# CC300), and the next day washed with 100 µL PBS per well and allowed to dry for at least 2 h prior to suspension cell seeding. Patient-derived whole blood was magnetically depleted of erythrocytes and granulocytes as above prior to labeling with CD45 magnetic beads and sorting into CD45− (Lin−) and CD45+ (Lin+) populations. After washing with PBS, cells were re-suspended in assay medium at a density of 1 × 10^5^ cells/100 µL per well into the collagen-coated plates and centrifuged for 1 min at 200× *g* with 0 braking. An amount of 80 µL of assay medium was then added to bring the final well volume to 180 µL. Cells were placed in a non-CO_2_ incubator at 37C for 1 h prior to running flux analysis.

The CTC-derived clone and MeWo control cells were plated at a density of 1 × 10^4^/100 µL in 96-well Seahorse Cell Culture Plates, and allowed to settle for one hour at room temperature before 100 µL of drug or placebo was added to yield final well concentrations of 1.5 µM palbociclib (LabShops, Cat# C1250-100mg, Claymont, DE, USA), 0.5 µM omacetaxine (ChemScene, Cat# 26833-87-4, Monmouth Junction, NJ, USA), both, or neither. Cells were then incubated in standard conditions for 3 days prior to flux analysis. For substrate oxidation experiments, 10 µM BPTES (Agilent, Cat# 103674-100, Santa Clara, CA, USA) or 10 µM UK5099 (Agilent, Cat# 103873-100) was added 24 h before flux analysis. For glutamine deprivation during drug treatment, DMEM without Phenol Red, D-glucose, L-glutamine, or sodium pyruvate (Gibco, Cat# A14430-01, Billings, MT, USA) was manually supplemented with 10 mM glucose (Agilent, Cat# 103577-100), 1 mM pyruvate (Agilent, Cat# 103578-100), 10% FBS, and 5 µM Hepes (Cytiva, Cat# SH30237.01, Marloboro MA, USA), with or without 2 mM glutamine (Agilent, Cat# 103579-100). We confirmed that this custom-supplemented media recapitulated the effects of our drug treatments compared to standard media, such that effects of glutamine deprivation were not due to the change in media formulation (Appendix A). Adherent cells were washed twice with 200 µL of supplemented assay medium and brought to a final volume of 180 µL per well prior to non-CO_2_ incubation. For adherent cells after 3 day drug treatment, flux data were normalized to MeanTsf values.

### 2.9. mRNA Isolation and RT-qPCR

Total mRNA was isolated using the RNeasy Micro Kit (Qiagen, Cat# 74004, Hilden, Germany) per manufacturer’s instructions. A maximum of 5 × 10^5^ cells were input for lysis and RNA extraction, and stored at −80 °C until ready for analysis. Amounts of 2 ng of total RNA per well were loaded into 0.1 mL MicroAMP fast 96-well Reaction Plates (Applied Biosystems, Cat# 4346907, Waltham, MA, USA) in 20 µL/well reactions and sealed with Optical Adhesive Covers (Applied Biosystems, Cat# 4360954). Each 20 µL reaction contained 5 µL of Taqman Fast Virus 1-step MasterMix (Thermo Fisher, Cat# 4444434), 1 µL of normalized RNA, 1 L of 20× Taqman Gene Expression Assays, and 13 µL of RNAse-free H_2_O supplied in the RNeasy Microkits. Taqman Gene Expression Assays were β-Actin (Hs01060665_g1), PGC-1α (Hs00173304_m1), PFKFB4 (Hs00894603_m1), GLUT1 (Hs00892681_m1), GLS (Hs01014020_m1), LDHA (Hs01378790_g1), LDHB (Hs00929956_m1), HIF1α (H Hs00153153_m1s), and PC (Hs01085875_g1). RT-qPCR was performed in duplicate with 2 ng/µL total RNA per well and following the cycling parameters specified for Fast Virus 1-step MasterMix for 40 cycles in a QuantStudio 6 Pro RT-PCR Analyzer, and final well ΔCT values were output by QuantStudio Version 6 software. Reported mRNA fold changes reflect 2^−ΔΔCT^ values. * *p* = 0.05, ** *p* = 0.01.

## 3. Results

### 3.1. The Translation Machinery and/or Cell Proliferation Are Critical for Successful CTC Micro/Macrometastases

First, 30 NBSGW mice were injected with an MBM CTC-derived clone. This CTC-derived melanoma clonal cell line was generated in Dr. Marchetti’s laboratory. It resulted from sequential selections (three) of cells injected intracardiacally in NSG mice, with the original clone being isolated from MBM. The CTC clonal cell line was shown to possess high-metastatic capabilities, found to consistently promote metastasis within two weeks from cell injection into animals, and to generate MBM as early as 24 h [28]. MBM mice were identified via IVIS imaging as early as 24 h post-injection. After one week, 20 MBM mice were equally divided into four cohorts (five mice in each cohort): placebo mice, mice treated with RPL inhibitor omacetaxine, mice treated with CDK4/CDK6 inhibitor palbociclib, and mice with combined drug treatment. Tumor progression was observed using IVIS which showed a higher tumor burden in placebo mice over time when compared to the drug-treated mouse groups (Figure 1A). Tridimensional reconstruction of tumor dissemination in all four animal groups was assessed using 3D IVIS tomography, showing downregulation of spatial metastasis in the drug-treated animals (Figure 1A). Notably, the MBM 3D signal was absent in the omacetaxine group with or without palbociclib. Total flux of intra- and extracranial metastases was quantified using IVIS which revealed a statistically significant decrease in signal in those mice that received individual or dual drug treatment (Figure 1B,C, with statistical analyses presented in Appendix A, respectively).

Whole body IVIS quantification of the total flux in the mouse revealed similar metastatic trends (Appendix A). Parallel animal necropsies confirmed a higher number of MBM mice and more extensive tumor burden in placebo mice (Figure 2A). A higher number of placebo mice developed extracranial macrometastases to distant organs, such as liver and bone marrow, when compared to drug-treated cohorts (Figure 2B). Micrometastasis quantification of tumor cells in mouse brains was consistent with macrometastatic analyses—placebo brains exhibited higher infiltration of tumor cells, when compared to brains from other groups (Figure 2C). Mouse sections of placebo brains had a 2–3-fold increase in tumor cells when compared to drug-treated brain sections (Figure 2D).

### 3.2. Inhibition of Translation and/or Cell Proliferation Decreased CTC Numbers and Altered Gene Signatures

To examine CTCs and changes in gene expression following drug treatment, we captured and analyzed CTCs from all animal groups. In each group, equal volumes of blood from three MBM CDXs were combined following retro-orbital blood draw and analyzed using the CTC Parsortix platform at the end of the study. Human CTCs were captured from mouse blood based on their size and deformability, and were then immunostained for human Mel-A Alexa Fluor 594, human FITC-CD45, and DAPI nuclear marker. CTCs were defined as Mel-A+/CD-45-/DAPI+ cells, visualized and quantified (Figure 3A and Table 1). Importantly, drug-treated CDXs exhibited a lower number of CTCs in their blood when compared to the positive control (Figure 3 and Table 1). Second, these observations were especially true for mice treated with omacetaxine with or without palbociclib. With omacetaxine treatment, the number of single CTCs decreased 4–6-fold in comparison with control (Figure 3 and Table 1). Third, blood from drug-treated mice did not contain any small or large CTC clusters. This finding is indicative of CTC decreased metastatic capacity and better therapy response in the drug-treated groups [61] (Figure 3A and Table 1). Additionally, CTCs from metastatic melanoma patients were captured and visualized using Parsortix (Figure 3B). As a negative control, blood of healthy donors was used to confirm the absence of CTCs (Figure 3C).

In parallel to Parsortix^TM^ staining, two blood samples from each animal group were processed by Parsortix to harvest CTCs for single-cell RNA-Seq analysis. RNA-Seq examination was performed to compare gene expression levels in different experimental groups, with libraries being aligned to the human genome. Hierarchical clustering revealed differential clustering of CTCs from placebo vs. drug-treated mouse groups (Figure 4, left panel). The displayed volcano plots show global gene expression of CTC populations with significant log2 fold change in the placebo samples versus the drug-treated samples (Figure 4, middle panel). Next, the differential gene expression between samples was generated using the Reactome^TM^ pathway database and resulted in the list of statistically significant pathways (Figure 4, right panel). Importantly, Reactome analyses of placebo versus omacetaxine groups generated an “Infectious disease” pathway that contains 41 RPL/RPS genes (Figure 4A, right panel—highlighted in yellow). Significantly, this pathway was present in the list of molecular pathways previously reported to be associated with the CTC RPL/RPS gene signature of MBM [28]. Another consequential pathway was “Metabolism of carbohydrates” which links RPL inhibition with metabolic alterations (Figure 4A, right panel—highlighted in red).

The second Reactome analysis against hallmark genes differentially expressed in placebo versus palbociclib generated a list of significant molecular pathways involved in the senescence and inhibition of cell proliferation (Figure 4B, right panel—highlighted in green). This result was expected since palbociclib inhibits CDK4/6 and therefore cell proliferation [62]. The third Reactome analysis between placebo and dual drug treatment produced a list of 35 upregulated significant pathways (Figure 4C, right panel). There were several important findings regarding this analysis. First, 34 out of 35 pathways had RPL/RPS genes and were involved in translational processes. Second, 33 out of 35 pathways were the exact signaling pathways related to the RPL/RPS gene signature of MBM [28]. Third, the only pathway that did not express any RPL/RPS genes was “Metabolism of carbohydrates” (highlighted in red). Fourth, the “Metabolism of carbohydrates” molecular pathway was found exclusively in omacetaxine-treated samples, with or without palbociclib (Figure 4A,C, right panels).

Furthermore, the mean value of each RPL/RPS gene in the MBM gene signature was evaluated. Notably, 4 of 21 RPL/RPS genes (*RPL23*, *RPL35A*, *RPL6*, and *RPS18*) were significantly downregulated in drug-treated groups when compared to placebo (Table 2). The fold-change difference in relation to placebo was quantified for each of the four members (Figure 5A). The crystal structure of the ribosome shows the protein configuration and highlights specific locations of RPL23, RPL35A, RPL6, and RPS18 proteins (Figure 5B). Strikingly, two of these four ribosomal proteins, RPL35A and RPL6, directly interact via hydrogen bonding, while the other two are found in opposite parts in the ribosome (Figure 5B). This could potentially indicate the important roles of these RPL/RPS proteins in ribosome assembly by being master regulators and recruiting other RPL/RPS members. Additionally, gene expression of proliferation markers was evaluated to determine palbociclib targets of proliferation (Appendix A). MYC, MYCL, MYCN, PCNA, E2F3, E2F4, E2F7, and E2F8 proliferation biomarkers were downregulated in CTCs from drug-treated mice, when compared to placebo. Notably, the proto-oncogene c-MYC has been reported to promote translational potential by increasing ribosome biogenesis [31,63].

### 3.3. Metabolic Profiling of CTCs

It has been previously reported that metabolic alterations can be associated with ribosome defects and thus can promote the oncogenic potential of ribosomal proteins [64,65]. Given that our Reactome analyses showed an altered carbohydrate metabolism after omacetaxine treatment (Figure 4A,C), alongside substantial decreases in CTC numbers in CDX models relative to placebo and palbociclib (Figure 3 and Table 1), we undertook in vitro real-time metabolic profiling of patient-derived neoplastic blood fractions. These were complemented with omacetaxine and/or palbociclib treatment of CTC-derived clones prior to metabolic flux analysis.

Real-time metabolic flux analysis via mitochondrial stress tests on Lin− and Lin+ cell fractions from patient-derived blood revealed substantial heterogeneity between patients. In the case of Seahorse^TM^ Patient 1, heterogeneity within the same patient with two months between blood collections was detected (longitudinal CTC monitoring) (Figure 6; patient parameters are listed in Appendix A). Two-way ANOVA of healthy donors and patient Lin− and Lin+ blood fractions showed all comparisons as non-significant (Figure 6B). However, the two-way unpaired Mann Whitney test revealed significantly elevated respiration in the patient Lin− fraction relative to the healthy donor Lin− fraction (Appendix A), an effect which persisted even after removal of Seahorse Patient 3′s exceptionally large and metabolically active Lin− fraction (Appendix A). Seahorse Patient 3′s Lin− fraction exhibited significantly elevated basal respiration, basal ECAR, and ATP-coupled respiration relative to the patient’s own Lin+ fraction and one of the healthy donor. Though the healthy donor’s Lin− fraction yielded only enough cells for two replicate wells which precludes statistical analysis, it was unambiguously distinct from the patient’s Lin− fraction (Figure 6C), suggesting an exceptional property of the circulating neoplastic cell fraction at the time of collection.

Three-day drug treatment of CTC clones with omacetaxine and/or palbociclib prior to metabolic flux analysis revealed strikingly similar results to those seen in CDX models: omacetaxine treatment resulted in substantial decreases in cell numbers, relative to both palbociclib- and placebo-treated cells (Figure 3, Table 1 and Figure 7A right panel). Real-time metabolic flux analysis via mitochondrial stress test (MST) and Glycolytic Rate Assay (GRA) revealed that CTC clones have lower baseline mitochondrial respiration and higher basal glycolysis than weakly metastatic MeWos (Appendix A). Additionally, CTC clones proved highly resistant to metabolic insult during flux analysis. They mounted far smaller compensatory responses to mitochondrial proton gradient decoupler FCCP during MSTs, and a smaller response to electron transport chain inhibition via Rot/AA during both GRAs and MSTs compared to weakly metastatic MeWos (Appendix A). However, the CTC-derived clone did respond robustly to the loss of glycolytic capacity in response to 2-DG injection (Figure 7C, left panel, and Appendix A, higher concentrations of mitochondrial proton gradient decoupler FCCP inhibitor not shown). Strikingly, cells retained a higher level of metabolic activity after palbociclib treatment than with omacetaxine, evidenced by higher per-cell OCR and ECAR after palbociclib treatment, and a decrease in both parameters after omacetaxine treatment (Figure 7 and Appendix A), bolstering Reactome analyses to show altered carbohydrate metabolism in this condition (Figure 4A,C).

Omacetaxine and/or palbociclib treatment alongside mitochondrial pyruvate transport inhibitor UK5099 revealed a greater degree of metabolic flexibility and competence remaining following palbociclib treatment. Only palbociclib-treated cells had excess pyruvate to direct toward aerobic glycolysis when mitochondrial transport was blocked by UK5099, suggesting that omacetaxine treatment left cells without spare fermentative capacity. While no treatment difference was significant in the presence of one-day treatment with glutaminase inhibitor BPTES, glutamine deprivation experiments alongside 3-day BPTES and inhibitor treatments revealed decreased cell counts in the absence of glutamine in both placebo- and palbociclib-treated cells, while omacetaxine treatment left cells largely without proliferative capacity in the presence or absence of glutamine (Appendix A).

From Reactome analyses of omacetaxine-treated cells, we selected key markers of carbohydrate metabolism that showed alteration relative to placebo-treated CDX mice and evaluated their expression at the mRNA level in CTC clones after inhibitor treatment by RT-qPCR. In omacetaxine-treated cells, two-way ANOVA revealed significant upregulation of *GLS*, *GLUT1*, and *HIF1α*, and significant downregulation of *LDHB*, *PC*, *PFKFB4*, and *PGC1α* relative to placebo-treated cells (Figure 7). Both omacetaxine and palbociclib showed substantially reduced oxidative capacity during glutamine deprivation, especially in the presence of 3-day treatment with BPTES (Appendix A), in line with prior research demonstrating glutamine’s flexible role in maintaining oxidative metabolism to sustain proliferation [66].

## 4. Discussion

This work provides first-time evidence that examining the melanoma CTC RPL/RPS gene signature and metabolic changes in patients’ CTCs is a critical prognostic factor of melanoma metastasis in general and MBM in particular. These findings were achieved by performing inhibitory drug studies in vivo and in vitro, in addition to metabolic profiling of a drug-treated CTC clone and the first-ever report of metabolic flux analysis on patient neoplastic cell fractions in real-time. Our previous work has employed a multilevel analysis to determine a common CTC signature for RPL/RPS genes which is linked to MBM onset and progression [28]. In this study, we targeted ribosomal biogenesis by using the translational inhibitor omacetaxine with or without the CDK4/CDK6 inhibitor palbociclib. Our results indicate that dysregulated ribogenesis inhibits MBM and progression of secondary metastasis. The identification of 4 out of 21 RPL/RPS genes downregulated following treatment further refines our approach to evaluate and prevent melanoma metastasis. Deregulation of ribosomal protein expression can accompany metabolic alterations in promoting cancer metastasis and resistance to therapy [31]. We expanded on these concepts by evaluating them in the context of metabolic flux and carbohydrate metabolism. Downregulation of the four RP subunits following omacetaxine treatment co-occurred with substantially decreased cell viability and proliferation as well as a quiescent metabolic state in vitro. Conversely, downregulation of the same four units in responses to palbociclib treatment reduced cell proliferation but did not result in reduced metabolic competency at the bulk flux level.

Each drug and the combination of drugs in this study were carefully selected based on specific mechanisms of action, and the important interplays between protein synthesis and cell-cycle progression [30,31,34]. Omacetaxine prevents the initial elongation step of translation by binding to the A-site cleft in the large ribosomal subunit [35]. However, the exact mechanism of inhibition is unknown. Palbociclib selectively inhibits cycLin−dependent kinases CDK 4/6 [38]. To our knowledge, this is the first reported evidence that four specific RPL/RPS genes (*RPL23*, *RPL35A*, *RPL6*, and *RPS18*) are downregulated in response to translation and proliferation inhibition (Table 2). Moreover, the 3D reconstruction of the ribosome showed the exact localization of these four RPL/RPS proteins (Figure 5B). Interestingly, two out of four RPL/RPS proteins, specifically RPL6 and RPL35A, are in direct proximity to each other and physically interact (Figure 5B). It is intriguing to speculate that these four RPL/RPS members act as master regulators of ribosome biosynthesis and they recruit other RPL/RPS proteins to finalize ribosome assembly. This work sets the stage for further mechanistic studies to examine the function of each RPL/RPS member and their binding partners.

Numerous studies have reported multiple cellular functions of the RPL/RPS proteins other than ribosomal assembly [31,64,67,68]. RPL/RPS extra-ribosomal functions include cell differentiation, cellular redox homeostasis, tumor suppression, cancer progression, immune signaling, p53 activation, oncogenic transformation, abnormal maturation of tumor cells, and DNA repair [64,68]. Changes in RPL/RPS expression can alter transcriptional-specific translational output, considering that upregulation of ribosomal proteins has been observed in cancer [69]. We have previously reported that the 21 RPL/RPS genes of the CTC signature are associated with melanoma brain metastasis development and progression [28].

Ribosome composition has been reported to be highly heterogeneous and dynamic [67], and different cell types have different ribosome composition due to having different binding partners [67,68,70]. Notably, deregulation of individual ribosomal proteins have specific patterns across cancer subtypes that can predict clinical outcomes [71,72] and specific somatic mutations in the RPL/RPS genes have been observed in various cancer types, including melanoma [64]. Importantly, oncogene-dependent cancer types have been shown to have enhanced activation of protein synthesis associated with metabolic rewiring of cells [73,74], and high ribosome biosynthesis is vital for proliferating and metastasizing cancer cells [75]. Disruption of ribosome biosynthesis results in ribosomopathies [64,76]. For example, deregulation of ribosomal protein synthesis via RPL15, and in turn protein translation, have been reported to be associated with breast cancer metastasis [30]. Alterations of RPL/RPS composition in cancer cells must be tightly regulated, since the imbalanced expression of RPL/RPS proteins can initiate the p53 response, leading to cell cycle arrest and apoptosis [64,68,77].

The diversity of RPL/RPS members and their specificity raise three intriguing possibilities for specific RPL/RPS contributions reducing the onset of MBM/extracranial metastasis: (1) the inhibition of the four RPL/RPS members (RPL22, RPL35A, RPL6, and RPS18) can alter the composition of ribosomes which may become specialized for the translation of specific oncogenic transcripts, similar to what has been observed in embryonic stem cells [78,79]; (2) the presence and action of extra-ribosomal RPL/RPS functions, e.g., the activation of p53-dependent/independent pathways in response to stress, resulting in cell cycle arrest and/or apoptosis [64]; (3) the involvement of RPL/RPS members in the translational control of circulating immune cells, affecting the development of metastasis. Examples are CTCs interacting with T cells where ribosomes lacking some RPs influence CD8+ T cell immunosurveillance [80], and neutrophils or myeloid-derived suppressor cells which are known to generate clusters with CTCs and whose translational capacity regulates the CTC metastatic propensities [25,81]. Collectively, these findings underscore the diversity of roles of RPL/RPS members in a diverse range of settings, connecting their canonical (“Ribogenesis”) and non-canonical functions (“Ribosomopathy”) with cancer.

Multiple oncogenes also known to modulate metabolic function, in particular MYC, have been linked to increased ribosomal synthesis and translation [29,63,82]. Our results demonstrate that targeting either cell ribogenesis machinery or cell proliferation results in a significant decrease in *MYC* and *MYCL* gene expression, along with gene expression of other proliferative biomarkers, e.g., *PCNA*, *E2F3*, *E2F4*, and *E2F8* (Appendix A). MYC controls transcription of many biomarkers, including RPL/RPS proteins [29,83], and c-MYC has been shown to promote expression of translational machinery, and ribosome and tRNA biogenesis directly [84]. Interestingly, alongside loss of *c-MYC* activity after inhibition of ribogenesis in the presence of both omacetaxine and palbociclib, only omacetaxine-treated CTCs showed upregulation of *HIF1α* at the mRNA level (Figure 7B). HIF1α and c-MYC exert overlapping actions to facilitate metabolic flexibility during metastatic progression, both complementary and antagonistic [84]. While both c-MYC and HIFα signaling can promote anabolic function directly and indirectly, HIFα does so by preventing glycolytic end-products from entering the mitochondria [84]. This elevation of *HIF1α* mRNA expression may represent a failed compensatory response to the loss of *c-MYC*’s anabolic functionality and overall anabolic incompetence during impaired ribogenesis, but not cell-cycle inhibition, as evidenced by the decreased basal glycolysis in omacetaxine but not palbociclib-treated cells. It is possible that upregulated mRNA expression of *GLS* and *GLUT1* represent part of this same failed compensatory response (Figure 7B).

Our results build on published studies since we show the link between proliferative inhibition and downregulation of RPL/RPS gene expression (Appendix A). Additionally, we show that the deregulation of carbohydrate metabolism that occurs alongside impaired ribogenesis represents a possible therapeutic vulnerability window of CTCs. Metabolic flux analysis on the brain metastatic CTC-derived clone showed that omacetaxine treatment substantially reduced its proliferation and viability, and pushed these cells toward a quiescent metabolic state (Figure 7A and Appendix A). While palbociclib also reduced metastasis in our CDX models, it caused elevation of both basal OCR and basal ECAR, and resulted in a smaller decrease in total CTC numbers in vitro and in vivo than did omacetaxine (Figure 3, Table 1, Figure 7A right panel, and Appendix A). It is possible that the anti-metastatic effects of palbociclib in vivo are independent of bulk metabolic flux alteration, or that altered patterns of interaction with CDX host cells and/or tissue microenvironments are required for this effect.

Given our results showing that omacetaxine treatment co-occurred with decreased metastasis and impaired carbohydrate metabolism in the CTC-derived clone, we suggest a high level of anabolic activity may occur in the presence or absence of elevated oxidative activity, and may represent a trait of metastasis-competent CTCs during dissemination. It is important to note that, due to the heterogeneity between tumors, CTCs derived from them, and the properties of derived metastatic tumors, the relatively lower level of oxidative activity in the MBM-competent CTC clone is independent of the High-OXPHOS and Low-OXPHOS MBM paradigm [40]. Taken together, metabolic flux and RT-qPCR analyses show that omacetaxine treatment of CTCs results in a decrease in metabolic flexibility relative to palbociclib, specifically the inability to re-direct pyruvate as a substrate to compensate for metabolic insult. Inhibitor treatment alongside mitochondrial pyruvate carrier (MPC) inhibitor UK5099 revealed a lack of excess pyruvate after omacetaxine treatment to re-direct from mitochondrial transport to glycolytic flux (Figure 7C). Downregulation of *PC* (pyruvate carboxylase), which yields oxaloacetate from pyruvate, suggests a loss of anaplerotic function after omacetaxine treatment.

Additionally, we found downregulation of *LDHB* but not *LDHA* after omacetaxine treatment (Figure 7B). *LDHA* preferentially converts pyruvate to lactate, while *LDHB* preferentially performs the reverse reaction, suggesting further loss of flexibility in using pyruvate as a substrate during impaired ribogenesis. *PFKFB4*, which confers anabolic flexibility by preferentially re-directing glucose to the pentose phosphate pathway [85,86] and enhances melanoma cell migration through RAS/AKT-dependent signaling independent of glycolysis [87], was also downregulated at the mRNA level after omacetaxine but not palbociclib treatment (Figure 7B).

Notably, we report the first-ever metabolic flux analysis in patient-derived CTCs in real-time. While there was considerable variation between and within patients, we saw overall elevated oxidative activity in patient Lin− compared to healthy donor Lin− samples (Appendix A), which persisted even after excluding the exceptional Lin− fraction of Seahorse Patient 3 (Appendix A). Seahorse Patient 3′s neoplastic fraction showed substantial elevation of both OCR and ECAR relative to comparable fractions (Figure 6C), and the extraordinary character of this patient’s Lin− fraction may reveal a rare snapshot of a particularly malignant CTC-shedding event.

The size of obtainable patient Lin− fractions may often preclude direct drug testing on patient-derived neoplastic cells during metabolic flux analysis, so examining direct effects of targeted inhibition on patient CTCs remains elusive. However, our findings of elevated oxidative metabolism in patient Lin− fractions and substantial elevation of metabolic activity (Seahorse Patient 3) highlight the interface of oxidative and anabolic metabolism as a potential therapeutic vulnerability during CTC dissemination. PC is a key mediator of anaplerosis in cancer cells and a central node for metabolic plasticity during cancer progression as it can promote both catabolic and anabolic metabolism [88]. PC is more highly expressed in cancer cells than in healthy cells; its overexpression is linked with the increased invasiveness of cancer cells and targeting PC with small molecule inhibitors has shown promise in suppressing cancer progression [89]. Additionally, inhibition of PC, along with mitochondrial pyruvate carriers (MPCs), has shown promise in sensitizing Tyrosine Kinase Inhibitor (TKI)-resistant leukemic stem cells to treatment [90]. We found that omacetaxine-treated CDXs show decreased CTC numbers and reduced MBM/extracranial metastasis, likely without the metabolic flexibility conferred by PC expression. Altogether, our findings suggest that disrupting PC function may be critical in clinical interventions to suppress and/or prevent metastasis.

Our results are altogether in line with previous findings showing heterogeneity in the oxidative capacity of neoplastic cells and the flexible use of carbohydrate metabolism to expand available metastatic niches [39,40,41,42,43,44]. We expand the current knowledge of CTC biology by demonstrating elevated oxidative metabolism in patient-derived Lin−/CTC populations relative to that of healthy donor blood. Employing the MBM CTC-derived clone, we demonstrate that impaired anabolic competency during dysregulated ribogenesis reveals a vulnerability for MBM-competent CTCs during dissemination. Targeting pyruvate metabolism can represent a potent and broadly viable therapeutic avenue for impairing CTC metastatic capacity, particularly in drug-resistant cancers. Future research characterizing PC expression in patient Lin−/CTCs at the mRNA/protein levels, in combination with PC expression assessment during glutamine deprivation, MPC inhibition, and metabolic flux analysis, will further elucidate how to leverage this CTC vulnerability for therapeutic efficacy.

We postulate that translationally activated, metastasis-competent CTCs could be novel therapeutic targets to combat cancer. Nevertheless, this study has some limitations. First, a limited number of melanoma patients was used in these analyses; therefore, we cannot claim that these observations are applicable to all melanoma patients. Heterogeneity between and within patients adds an additional level of complexity in the way of reaching definite conclusions, and further studies are warranted. Second, the drug inhibitory investigations used a limited number of CDX mice, which, although statistically significant, may introduce inherent sampling bias. Third, because we did not execute an inhibitory study in humanized mice, we cannot exclude the possibility that the drugs could affect interactions of CTCs with circulatory immune cells that could alter the immune response. Fourth, while our CTC enrichment method removes RBCs, PBMCs, granulocytes, and platelets upstream of real-time metabolic flux analysis, our current knowledge of CTC biology suggests that it is unlikely that the entirety of collected Lin−/CTC fractions are all CTCs per se. Nevertheless, marker-agnostic approaches to neoplastic cell characterization are necessary to fully capture the entire spectrum of metastasis-competent CTCs [43,91]. Our analyses indicated a trend toward greater size of patient Lin−/CTC fractions (Appendix A), as well as discrete metabolic characteristics of sorted Lin−/CTC fractions from patients relative to those from healthy donor blood (Figure 6C, Appendix A). We therefore suggest that our analyses reveal unique metabolic properties of melanoma patient Lin−/CTC fractions. Fifth, bulk flux analysis of CTCs after drug treatment may miss subtler shifts in metabolic state. While our targeted, Reactome-driven examination of the critical dimensions of carbohydrate metabolism demonstrated the power of this approach for rigorously characterizing drug responses, future functional assays alongside proteomic and metabolomic approaches will more fully characterize CTC metabolic vulnerabilities during targeted therapeutic disruption of the onco-ribosome, as well as revealing further heterogeneity of CTC metabolism [42,43,91,92]. Palbociclib’s anti-metastatic effects in vivo show that the loss of metabolic flexibility accompanying impaired carbohydrate metabolism downstream of disrupted ribogenesis represents only one possible pathway to preventing metastasis. However, omacetaxine yielded greater cell death, and decreased metabolic competency and flexibility in vitro, as well as fewer CTCs in vivo, than did treatment with palbociclib, which may be critical for complete elimination of rare subpopulations of metastasis-competent CTC before and/or during dissemination.

## 5. Conclusions

This study represents pioneering work that identifies the relevance of targeting the CTC RPL/RPS signature to suppress melanoma metastasis in general and MBM in particular. The systemic examination of patients’ RPL/RPS gene expression in CTCs could be pivotal to prescribing more targeted treatment based on the needs of the patient. Future experimental and analytical innovations built on our findings will aid in novel therapeutic interventions and improvement of clinical outcomes.

## Figures and Tables

**Figure 1 cancers-15-05263-f001:**
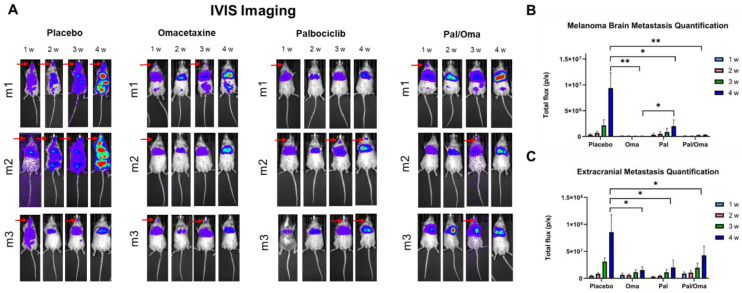
IVIS imaging and analysis after treating CTC-derived clone-injected MBM mice with the RPL inhibitor (omacetaxine), the CDK4/6 inhibitor (palbociclib), or the combination of drugs. Placebo group was used as a positive control. (**A**) tumor burden of each mouse was monitored via IVIS imaging weekly. Three representative mice in each group are shown (5 mice in each cohort). IVIS quantification of intra- (**B**) or extracranial (**C**) mouse regions shows decrease in tumor burden in drug-treated groups, compared to placebo. See “Materials and Methods” for experimental details, and Appendix A for statistical analyses. * *p* = 0.05, ** *p* = 0.01.

**Figure 2 cancers-15-05263-f002:**
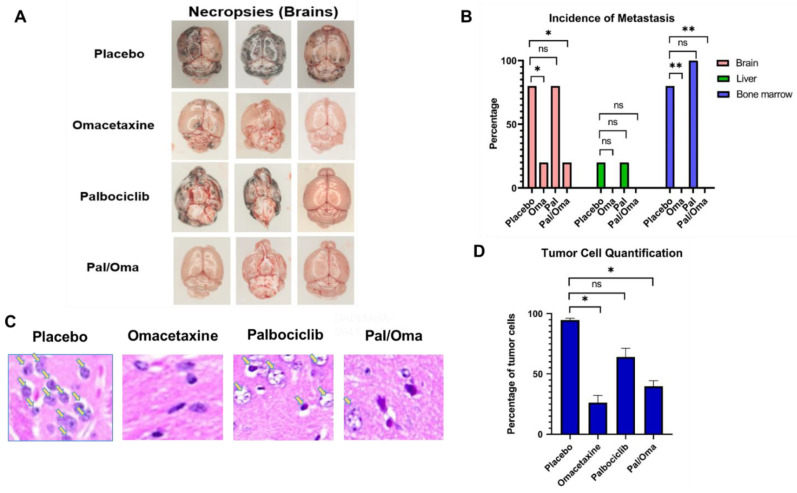
CTC-driven micro- and macrometastasis quantification. (**A**) representative mouse brains from four animal groups are shown. (**B**) Bar graphs show the percentage of macrometastases from distant sites. Incidence of metastasis was found to be statistically significant between placebo- and drug-treated groups. (**C**) Melanoma cells in the brain sections from four animal groups using H&E staining (yellow arrows). (**D**) Quantification of tumor cells in the brain sections. See “Materials and Methods” for experimental details. ns = non-significant, * *p* = 0.05, ** *p* = 0.01.

**Figure 3 cancers-15-05263-f003:**
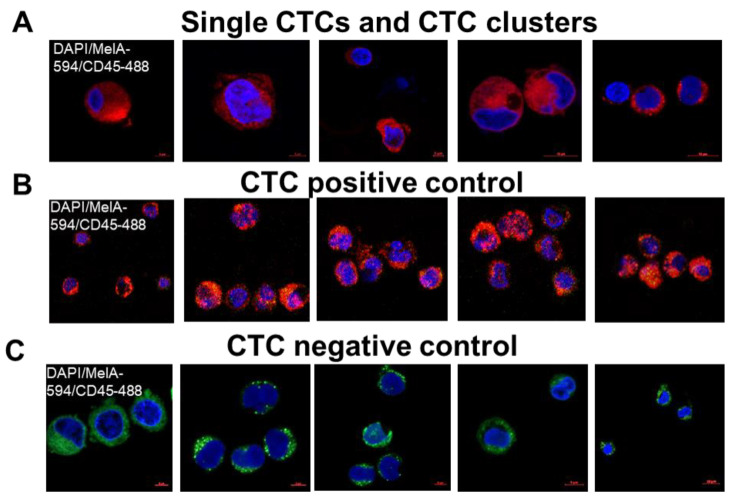
Enumeration and examination of CTCs captured by the Parsortix platform. Red box shows enumeration of large CTC clusters in each group. Visualization of CTCs and CTC clusters in mouse blood (**A**), melanoma patients’ blood (**B**), or healthy donor blood (**C**). See “Materials and Methods” for experimental details.

**Figure 4 cancers-15-05263-f004:**
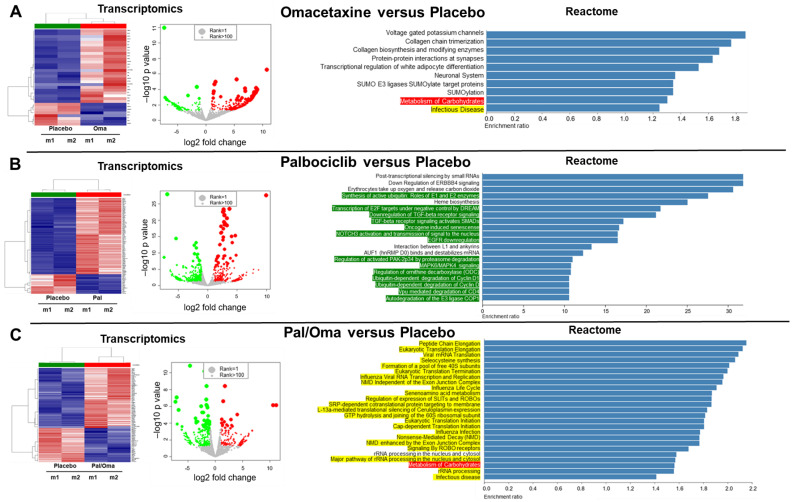
RNA-Seq analyses of Parsortix-harvested CTC-enriched cell populations in four different mouse groups. Mouse blood from each experimental group was collected, pooled, and analyzed using Parsortix. Cellular fractions enriched with CTCs were harvested, followed by RNA isolation and transcriptomic profiling. Hierarchical clustering of gene expression profiling showed differential clustering of CTCs from placebo mice versus CTCs from omacetaxine-treated mice (**A**), CTCs from placebo mice versus CTCs from palbociclib-treated mice (**B**), or CTCs from placebo mice versus CTCs from mice receiving combinatory drug treatment (**C**). Corresponding heat maps are shown (left panels, red = upregulated, blue = downregulated). Scatter plots display global gene ex-pression of CTC populations with significant log2 fold change in the placebo group (green dots) versus the drug-treated groups (red dots; middle panels). Right panels show statistically significant pathways resulting from the hierarchical clustering using Reactome analysis. Pathways highlighted in red are the molecular pathways related to metabolic profiling. Highlighted in yellow are the pathways containing the CTC RPL/RPS gene signature of MBM [28]. Molecular pathways associated with proliferation are highlighted in green. See “Materials and Methods” for experimental details.

**Figure 5 cancers-15-05263-f005:**
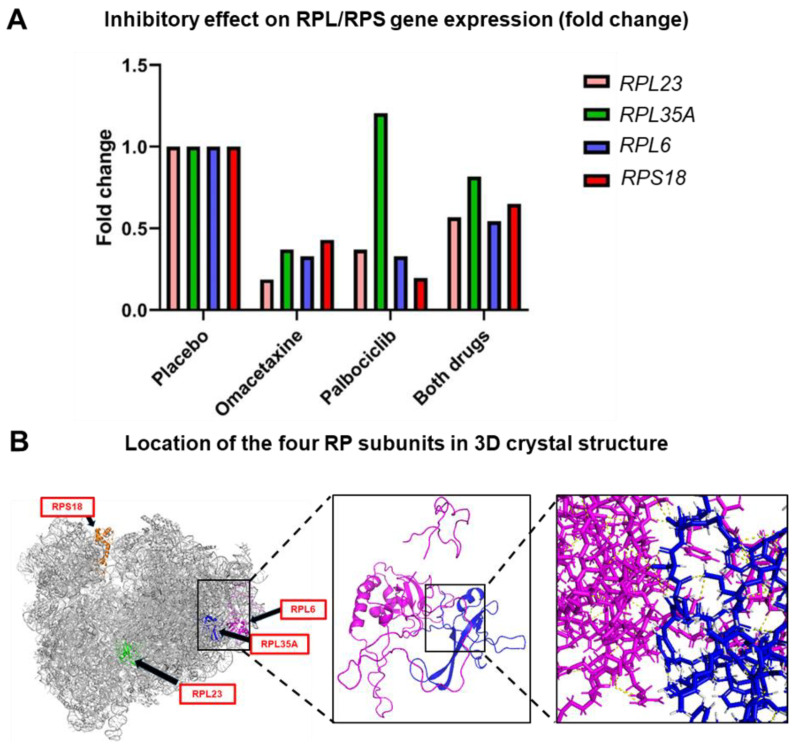
Four RPL/RPS genes (RPL23, RPL35A, RPL6, and RPS18) were inhibited by drug treatment. (**A**) representation of fold-change difference between drug-treated groups and placebo. (**B**) The 3D crystal structure of a ribosome generated by the PyMOL software shows specific localization of RPL23, RPL35A, RPL6, and RPS18 (black arrows). The left two panels show direct physical interaction between RPL6 (highlighted in purple) and RPL35A (highlighted in blue) via hydrogen bonding.

**Figure 6 cancers-15-05263-f006:**
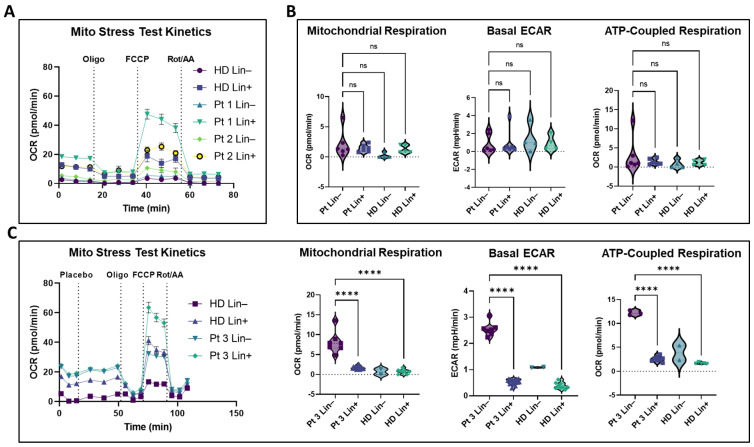
Metabolic flux in patient-derived Lin−/CTC fractions. (**A**) Representative Mito Stress Test kinetics after lineage separation of two patients and a healthy donor sample. (**B**) Pooled metabolic parameters for the entire patient dataset. (**C**) OCR kinetics and metabolic parameters of Patient 1, showing basal respiration, basal ECAR, and ATP-coupled respiration to be elevated in the patient Lin−/CTC fraction. Violin plots all represent normalized data. All analysis was performed using two-way ANOVA with Kruskal–Wallis multiple comparisons test. OCR parameters at or below background threshold that showed appropriate well kinetics were treated as “0” values. Basal ECAR from Mito Stress Tests represent per-well averages of the three ECAR reads preceding the first drug injection. Analyzed patient samples all displayed robust flux kinetics and all experiments included control healthy donor cells on the same Seahorse plate. ns = non-significant, **** *p* < 0.0001.

**Figure 7 cancers-15-05263-f007:**
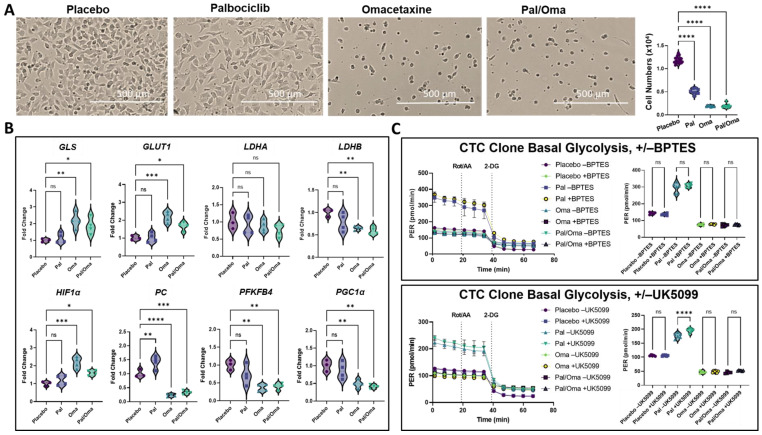
Cell numbers and carbohydrate metabolism in CTC-derived clone after inhibitor treatment. (**A**) Representative images showing cell density and morphology after treatment, as well as cell counts (top right panel) using a Biotek Cytation 1 with Gen 5 3.11 software after metabolic flux analysis. (**B**) Fold-change RT-qPCR analyses, and analysis using two-way ANOVA with Kruskal–Wallis multiple comparison test. (**C**) Glycolytic Rate Assay kinetics during 3-day inhibitor treatment in the presence or absence of 10 µM BPTES (top panel) or 10 µM UK5099 (bottom panel) for the last 24 h of inhibitor treatment. ns = non-significant, ** p <* 0.05, *** p <* 0.01, **** p <* 0.001, ***** p* < 0.0001.

**Table 1 cancers-15-05263-t001:** Parsortix CTC quantification from CDXs.

CTC Clusters	Placebo	Omacetaxine	Palbociclib	Pal/Oma
	CTCs/mL			
Single-cell	60	15	35	10
2-cell	20	0	10	0
3-cell	10	0	0	0
4-cell	10	0	0	0
5-cell or greater	5	0	0	0

**Table 2 cancers-15-05263-t002:** Inhibitory effect on RPL/RPS gene expression (cpm value).

	Placebo	Omacetaxine	Palbociclib	Both Drugs	CTC Clone
RPL12	2.14	1.71	4.81	7.95	11.4
RPL13	5.1	3.14	5.55	3.2	177.59
RPL18A	1	2.43	1.77	1.73	16.78
RPL19	81.1	108.64	154.07	143.33	43.91
RPL26	3.29	1.71	3.27	4.35	6.5
RPL37	15.35	15.98	22.96	6.72	73.96
RPL38	1.91	3.85	5.6	2.39	27.73
RPL7	3.29	2.42	1	1.73	15.61
RPL7A	8.72	5.27	1	1	46.48
**RPS18**	**8.96**	**3.85**	**1.75**	**5.82**	**22.42**
**RPL6**	**3.05**	**1**	**1**	**1.65**	**7.07**
**RPL35A**	**4.62**	**1.71**	**5.57**	**3.78**	**69.09**
**RPL23**	**13.01**	**2.43**	**4.81**	**7.37**	**35.69**
RPS12	60.17	48.06	33.63	81.2	17.56
RPS15A	1	1	1	1	20.35
RPS24	1	1	1.75	4.35	24.82
RPS26	1	1	1	1	3.51
RPS28	1	1	1	1	12.91
RPS5	5.1	1	2.54	9.42	15.37
RPS7	1.91	1.71	2.5	3.12	5.03
RPSA	12.39	4.56	7.09	20.05	28.92
Average	11.2	10.17	12.56	14.87	32.51

## Data Availability

Not applicable.

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
