# Peer review of "Targeting Translation and the Cell Cycle Inversely Affects CTC Metabolism but Not Metastasis"

_cancers, 2023, doi:10.3390/cancers15215263_

Round 1

Reviewer 1 Report

Comments and Suggestions for Authors

This manuscript titled ‘Targeting translation and the cell-cycle inversely affects CTC metabolism but not metastasis’ is a very interesting piece of work.

It reports the critical issue of melanoma brain metastasis (MBM), which carries a grim prognosis and is frequently diagnosed at advanced stages. By investigating the role of CTCs and their association with MBM, the researchers propose a nice approach targeting ribogenesis to prevent metastasis in CTC-derived xenografts. They employed FDA-approved inhibitors, omacetaxine, and palbociclib, which demonstrate promising outcomes in reducing MBM and extracranial metastasis. The downregulation of specific RPL/RPS genes in mouse blood-derived CTCs suggests a potential mechanism.

However, the conflicting effects on glycolytic metabolism raise questions about the precise mechanisms at play. Additionally, characterizing patient-derived CTCs provides valuable insights into their metabolic functions. The study makes a significant contribution to understanding MBM and CTC biology, but further research is needed to elucidate the metabolic intricacies and clinical translation of these findings.

I have few queries:

1.       Please provide a good schematic of the work which highlights the core of the work.

2.       Please reorganise the figure 4 for better clarity.

3.       Authors have identified specific RPL/RPS genes which were downregulated in mouse blood-derived CTCs. How do these genes contribute to the reduced development of MBM and extracranial metastasis? More details will be appreciated.

4.       How might the metabolic characterization of patient-derived circulating neoplastic cells/CTCs be leveraged for clinical applications in diagnosing, treating for other tumor cells. What are the potential challenges in translating these findings into practical therapies or interventions for patients?

Reviewer 2 Report

Comments and Suggestions for Authors

Here the authors investigated Targeting translation and the cell-cycle inversely affects CTC metabolism but not metastasis. They make the interesting observation that CDK4/6 inhibitor decreased MBM/extracranical metastasis and real time metabolic flux analysis on patient derived melanoma CTCs, and altered carbohydrate metabolism during impaired translation in a melanoma CTC derived clone. 

This reviewer likes the manuscript as it provides new discoveries. However, some sections in the manuscript need to be backed up by more solid evidence from statistical analysis. 

Major points

1. Please perform statistical analysis in Figure 1C and D, then state your conclusion based on your statistical analysis. Generally, put 5 mice data into 1 cohort (i.e. placebo), then perform statistical analysis. 

2. Please perform statistical analysis in Figure 2B and D, then state your conclusion based on your statistical analysis.

Major points

1.    Please add CTC-derived clone information in Results 3.1, although there are in materials and methods because CTC-derived clone is unfamiliar to readers.

2.    Please enlarge Figure 1B, because it is hard to see brain metastasis.

Round 2

Reviewer 2 Report

Comments and Suggestions for Authors

I am satisfied with the revisions that have been made by authors.